# Public sentiment and thematic evolution in the metaverse: A large-scale computational analysis of Twitter discourse

Samuel Duraivel[1☉], Lavanya Rajendran[2☉], Srinidhi Vasudevan[3☉*], Anna Piazza[3☉]

1 Research and Development, Jubilee Mission Group of Institutions, Bangalore, India, 2 Department of Media Sciences, College of Engineering, Guindy, Anna University, Chennai, India, 3 School of Business, Operations and Strategy, University of Greenwich, London, United Kingdom

☉ These authors contributed equally to this work.
* srinidhi.vasudevan@greenwichac.uk

## Abstract

This study investigates public perception of the metaverse through a large-scale computational analysis of 52,874 English-language tweets. Leveraging sentiment analysis tools (VADER and RoBERTa) and unsupervised topic modeling (BERTopic), we categorize discourse into four thematic domains: general metaverse discussion, Meta's Horizon Worlds, metaverse-related cryptocurrency tokens, and virtual social events. Our findings reveal that 43.0% of tweets express positive sentiment, driven by enthusiasm for immersive innovation and digital transformation, while 23.6% convey skepticism, primarily concerning platform reliability, corporate dominance, and privacy. Sentiment surrounding Horizon Worlds reflects a paradox: underlying optimism is overshadowed by user frustration, with negative tweets generating disproportionately high engagement. Analysis of metaverse token discourse indicates robust investor interest, tempered by persistent concerns over market volatility and fraudulent schemes. Topic modeling further uncovers a notable narrative shift from speculative price-focused discussions toward utility-driven use cases. Virtual events (e.g., digital weddings, concerts) elicit the most positive sentiment (51.3%), with users frequently expressing emotional resonance and communal belonging, as visually reinforced by word cloud analysis. This research contributes to the literature on digital adoption and emerging technologies by mapping the evolving social discourse of the metaverse. It offers actionable insights for platform developers, investors, and educators seeking to align innovation with user expectations and provides a predictive lens for forecasting public readiness for the next generation of digital interaction.

## 1. Introduction

The metaverse, a collective virtual shared space, created by the convergence of virtually enhanced physical reality and physically persistent virtual space, has emerged

---

**Data availability statement:** All relevant data are within the manuscript and its Supporting information files.

**Funding:** The author(s) received no specific funding for this work.

**Competing interests:** The authors have declared that no competing interests exist.

as one of the most contested and promising frontiers of digital innovation [1]. While corporate entities such as Meta (formerly Facebook), Microsoft, and NVIDIA have invested billions into its infrastructure [2], public reception remains fragmented, oscillating between utopian enthusiasm and dystopian skepticism [3]. This polarization is not merely rhetorical; it reflects deeper tensions around data sovereignty, digital labor, and the corporatization of virtual public spheres [4]. Understanding this public sentiment is not merely an academic exercise; it is a strategic imperative for developers, policymakers, and investors navigating the uncharted terrain of Web 3.0 [5]. Recent empirical studies indicate that these technologies are fundamentally transforming various virtual environments, reshaping user interaction models across educational, social, and professional domains [1,2]. *Yet despite this institutional investment, broader public adoption remains constrained by systemic security, privacy, and trust challenges, factors increasingly recognized as critical prerequisites for sustainable metaverse deployment [6]. Understanding how these structural barriers intersect with lived user experience, as expressed in organic public discourse, is therefore essential for developers, policymakers, and investors navigating this terrain.*

Social media platforms, particularly Twitter, serve as real-time barometers of public opinion [7]. Unlike surveys or focus groups, which impose researcher-defined frames, Twitter data captures organic, unprompted expressions of sentiment, a critical advantage when studying emergent technologies whose cultural implications are still being negotiated [8]. Prior studies have successfully leveraged Twitter to track public responses to AI [9], vaccine hesitancy [10], and cryptocurrency adoption [8], demonstrating its utility as a proxy for collective effect. This study builds on that tradition by analyzing 52,874 English-language tweets spanning a six-month period (January 1, 2023 to June 30, 2023), employing state-of-the-art computational methods to decode the thematic structure and emotional valence of metaverse-related discourse.

Our research addresses three core questions: What are the dominant thematic clusters within public metaverse discourse? How does sentiment vary across thematic domains and over time? And which topics generate the highest user engagement, and what does this imply for platform development and investment strategy? To answer these, we deploy a dual-method sentiment analysis framework (VADER and fine-tuned RoBERTa) and BERTopic for unsupervised thematic modeling, methodologies selected for their complementary strengths in handling social media's noisy, context-dependent language [11]. Our findings reveal not only the emotional temperature of the metaverse conversation but also its latent narrative arcs, from speculative hype to experiential utility, offering a predictive model for future adoption curves grounded in real-world discourse dynamics [12].

This paper proceeds as follows: Sect 2 reviews relevant literature; Sect 3 details methodology with expanded technical specifications; Sect 4 presents results with integrated tables and figures; Sect 5 discusses implications in theoretical and practical context; and Sect 6 concludes with limitations and future directions.

## 2. Literature review

### 2.1 The metaverse: Definitions and debates

The term "metaverse" was first coined by Neal Stephenson in his 1992 novel Snow Crash, describing a fully immersive 3D virtual world populated by user-controlled avatars [13]. Contemporary definitions vary, but most converge on a persistent, interoperable, multi-user virtual environment enabled by VR/AR, blockchain, AI, and real-time rendering [14]. Some scholars frame the metaverse as an extension of existing digital ecosystems [15], while others argue it represents a paradigmatic rupture, a "spatial internet" that reconfigures human presence and interaction [16]. Critics contend that current implementations remain technologically immature and conceptually incoherent, amounting to little more than rebranded VR platforms [17]. Proponents, however, see it as the inevitable next layer of human-computer interaction, a post-mobile computing environment where digital and physical realities converge [18]. This tension between speculative promise and grounded critique forms the backdrop of our analysis, as public discourse often mirrors these polarized expert positions [19].

### 2.2 Sentiment analysis in social media research

Sentiment analysis has become a cornerstone of computational social science, particularly in the context of emerging technologies where traditional survey methods lag behind real-time public opinion [20]. Lexicon-based tools like VADER (Valence Aware Dictionary and sEntiment Reasoner) excel in social media contexts due to their sensitivity to slang, emojis, and capitalization, features that often confound traditional NLP models [21]. Transformer-based models like RoBERTa offer superior contextual understanding by capturing long-range syntactic and semantic dependencies [22], but they require fine-tuning for domain specificity to avoid misclassification of neologisms and platform-specific jargon [23]. Recent work has demonstrated that ensemble approaches combining lexicon-based and transformer models significantly improve classification accuracy in noisy social media corpora [24]. Following this precedent, we adopt a dual-model architecture to ensure robustness and mitigate algorithmic bias, a methodological choice further validated by comparative studies in crisis communication analysis [25].

### 2.3 Topic modeling and public discourse mapping

Latent Dirichlet Allocation (LDA) dominated early topic modeling, but its bag-of-words assumption limits semantic coherence, particularly when modeling discourse around evolving technologies where word meaning shifts rapidly [26]. BERTopic, which leverages sentence transformers and class-based TF-IDF (c-TF-IDF), generates more interpretable and contextually rich topics by preserving semantic relationships between phrases [27]. This approach has been successfully applied to map vaccine hesitancy narratives [10], climate change framing [28], and cryptocurrency sentiment [29], demonstrating its versatility in capturing emergent themes in noisy, high-velocity social data. Its ability to dynamically adjust cluster granularity via HDBSCAN makes it particularly well-suited for mapping the conceptual evolution of metaverse discourse from abstract speculation to concrete use cases without imposing rigid pre-defined categories [30].

### 2.4 Gaps in current research

Despite growing scholarly interest, few studies have systematically mapped public sentiment across multiple thematic domains of the metaverse. Most focus narrowly on investment trends [31] or technical architectures [32]. A 2022 review noted the "striking absence of large-scale empirical studies on user perception" beyond controlled lab environments [33]. None, to our knowledge, have analyzed engagement metrics (likes, retweets, replies) as a function of sentiment polarity, a critical oversight, given that negative content often drives virality due to evolved psychological biases toward threat detection [34]. This study fills these gaps by offering: (1) a multi-domain sentiment taxonomy grounded in unsupervised clustering, (2) temporal analysis of thematic evolution using rolling-window sentiment scoring, and (3) engagement-weighted

sentiment profiling with multivariate regression controls, all derived from a large, real-world dataset collected during a period of peak public interest and media scrutiny.

## 3. Methodology

### 3.1 Data collection and sampling strategy

We collected a total of 52,874 English-language tweets through a hybrid data collection strategy necessitated by platform access changes during the study period (January 1 to June 30, 2023), Initially, data was queried using the **Twitter API v2 Academic Research track**. Following changes to API access tiers that restricted historical archive querying, we completed the longitudinal dataset using **snscrape**, a Python-based social media scraping tool widely utilized in computational social science to ensure continuity in temporal data. Both collection methods utilized an identical set of search keywords and hashtags to ensure consistency: "metaverse," "#metaverse," "Horizon Worlds," "metaverse token," "*SAND*," "MANA," "$AXS," "virtual wedding," "metaverse concert," and "digital event". The resulting raw corpus was homogenized to remove collection artifacts specific to either method. All tweets were deduplicated using exact text matching and SHA-256 hashing, filtered for non-English content using the fastText language identification model (threshold >0.9 confidence), and preprocessed by removing URLs, user mentions, and non-ASCII special characters while preserving emojis for sentiment fidelity. This dual-method approach ensured continuous temporal coverage across the six-month window despite infrastructural disruptions, capturing the full evolution of discourse from speculative token economics to the technical realities of Horizon Worlds

**Data cleaning and bot mitigation:** To ensure the analysis captured organic public discourse rather than automated amplification, a strict deduplication protocol was applied. All retweets and duplicate text strings were removed using SHA-256 hashing, resulting in a dataset exclusively composed of unique narrative instances (n = 52,874). While a dedicated machine-learning bot classifier was not employed, this rigorous deduplication served as an effective heuristic filter for high-volume automated spam, which typically relies on repetitive text dissemination. Furthermore, manual inspection of the resulting topic clusters (see S1 File) revealed coherent semantic structures devoid of the disjointed syntax characteristic of algorithmic content generation. Retweets were therefore excluded from the textual sentiment analysis to prevent inflationary bias, though their volume was captured in the weighted 'Engagement Score' to reflect content visibility.

### 3.2 Sentiment analysis framework: Model selection and validation

To capture the emotional nuance of metaverse discourse, we employed a hybrid sentiment analysis framework that integrates the rule-based VADER lexicon with a fine-tuned RoBERTa transformer model. We selected VADER for its computational efficiency in handling social media vernacular, such as emojis and capitalization, while RoBERTa was chosen for its superior ability to detect context-dependent sentiment in complex sentences. To ensure the validity of this hybrid approach, we constructed a ground-truth validation dataset (S2 File, n = 5,000) consisting of tweets labeled by human annotators. The RoBERTa model was fine-tuned on a composite corpus and achieved an F1-score of 89.2% on the held-out test set, demonstrating high classification accuracy. The annotation guidelines followed a three-category schema (Positive, Neutral, Negative) with clear operational definitions: Positive tweets expressed enthusiasm, endorsement, or optimism about the metaverse; Negative tweets conveyed criticism, frustration, or skepticism; Neutral tweets provided factual information without clear emotional valence. Annotators were instructed to prioritize contextual sentiment over lexical cues alone, particularly for sarcasm and irony.

**Inter-model agreement and error analysis:** To transparently evaluate the hybrid architecture, we constructed a confusion matrix (Fig 1) comparing VADER's lexicon-based predictions against the fine-tuned RoBERTa model on the validation set (n = 5,000). The matrix reveals that the majority of disagreements (7.8%) stemmed from VADER's tendency to classify subtle, context-dependent positive sentiment as 'Neutral,' a limitation inherent to lexicon-based approaches that

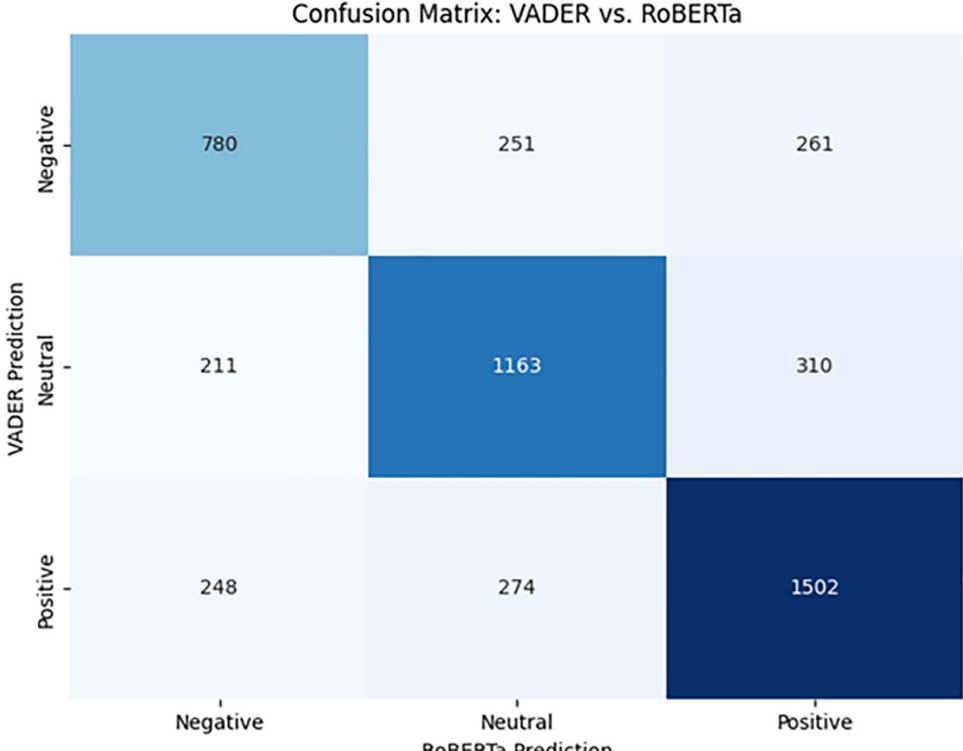

**Fig 1. Confusion Matrix of VADER vs. RoBERTa Predictions (n = 5,000).** Diagonal cells represent agreement; off-diagonal cells indicate misclassifications. The 7.8% disagreement rate primarily reflects VADER's tendency to classify context-dependent positive sentiment as neutral.

lack attention mechanisms. RoBERTa successfully captured these nuances, justifying its higher weight in the ensemble decision logic.

We applied a majority-vote ensemble method to assign final sentiment labels; in instances where VADER and RoBERTa diverged—approximately 7.8% of the dataset as recorded in the validation file—a human adjudicator resolved the disagreement to maintain high inter-rater reliability (Cohen's κ = 0.82). This "human-in-the-loop" protocol ensured that the high negativity observed in technical domains was a genuine reflection of user frustration rather than algorithmic artifact.

### 3.3 Topic modeling with BERTopic: From micro-topics to macro-themes

We utilized BERTopic to uncover the latent semantic structure of the dataset, selecting it for its ability to generate dense clusters from noisy social media text using sentence transformers and c-TF-IDF class-based weighting. The algorithm identified a stable solution of 28 distinct micro-topics, the full taxonomy of which is provided in S1 File to facilitate reproducibility. These micro-topics capture highly specific community concerns, ranging from "VR Headset Issues" (Topic 2) and "Audio/Voice Glitches" (Topic 6) to "Price Speculation" (Topic 14) and "Virtual Weddings" (Topic 7). To enable the high-level statistical comparisons presented in our results, we aggregated these 28 micro-topics into four verified macro-themes based on semantic proximity and domain expert review: General Discourse, Horizon Worlds, Metaverse Tokens, and Virtual Events. For example, the high negativity reported in the "Horizon Worlds" macro-theme is directly attributable to the aggregation of technical complaint clusters such as "Bugs" (Topic 0) and "User Safety" (Topic 4), whereas the "Metaverse Tokens" theme aggregates economic subtopics like "NFT Drops" (Topic 16) and "Gas Fees" (Topic 19).

## 3.4 Engagement metrics, statistical modeling, and robustness checks

User engagement was quantified using a weighted Engagement Score (ES) calculated via the formula:

$$ES = Likes + (1.5 \times Retweets) + (2 \times Replies) + (3 \times QuoteTweets)$$

These weights were selected to reflect the increasing cognitive effort and network visibility associated with each interaction type [35,36]. A Like represents a low-cost signal requiring minimal cognitive processing and generating limited visibility beyond the original author's notifications [37]. A Retweet (weighted 1.5×) amplifies content to the retweeter's follower network and functions as an endorsement signal, involving greater intentionality and reputational exposure [38, 39]. A Reply (weighted 2×) requires users to generate original content, demanding higher cognitive engagement and creating a publicly visible conversational thread [39,40]. A Quote Tweet (weighted 3×) entails the highest editorial effort, as users must both interpret the original content and contribute additional commentary, while simultaneously increasing network visibility through diffusion across both the quoter's and original author's audiences [39]. To account for the heavy-tailed distribution characteristic of social media engagement data—where a small number of viral tweets skew the arithmetic mean—we report the **Median (Mdn)** and **Inter-quartile Range (IQR)** alongside means to provide a more robust measure of central tendency. Statistical significance was assessed using non-parametric tests (Kruskal-Wallis H test) where normality assumptions were violated, alongside multivariate linear regression to isolate the effects of sentiment and thematic content on engagement intensity.

## 3.5 Acknowledging data source limitations

While Twitter provides valuable real-time discourse data, several representativeness limitations must be acknowledged. Twitter's user base exhibits demographic skews that may affect generalizability, as its users tend to be younger, more educated, more urban, and more politically engaged than the general population [41]. Internationally, Twitter usage varies significantly, with higher penetration in English-speaking countries and lower adoption in regions where metaverse development is actively occurring (e.g., South Korea, Singapore). These demographic patterns may systematically bias our sentiment measures toward tech-savvy, English-speaking, and Western perspectives. Furthermore, the absence of granular demographic data (such as gender and geographic region) in our dataset limits our ability to analyze how sentiment varies across different population segments. Prior research has established that metaverse perception is often influenced by demographic factors, with distinct variations observed across gender lines [42] and regional adoption rates [43], highlighting the importance of diversity-aware analyses in virtual environment studies. Consequently, while our findings map the broader digital discourse, they should not be interpreted as a demographically stratified representation of global public opinion. Additionally, Twitter's character limit and public nature may encourage more polarized expressions compared to private conversations or longer-form content. Established research indicates that sentiment expressed on Twitter can exhibit greater emotional intensity than measures from other platforms or surveys [44]. To partially address these limitations, we: (1) focus our interpretations specifically on "Twitter-based public discourse" rather than claiming broader population representativeness, and (2) acknowledge that our results may not generalize to non-Twitter users or private sentiment. Future work should incorporate platform-diverse datasets (e.g., Reddit, Discord, TikTok) and leverage demographic proxy methods or survey linkage to stratify sentiment by age, gender, and geographic region, as demonstrated in recent computational social science studies [42,43].

## 4. Results

### 4.1 Overall sentiment distribution and representativeness

The final analytical corpus consisted of 52,874 unique, original tweets. By excluding retweets from the sentiment calculation, we ensured that the reported distribution—43.0% positive, **33.4%** neutral, and 23.6% negative—reflects the diversity

of distinct user viewpoints rather than the viral amplification of a few dominant narratives. This distinction is critical for interpreting the 'Horizon Worlds' sentiment, where negative technical reports were unique, individual complaints rather than a single viral post retweeted by bots. Representative positive tweets included expressions such as "The metaverse is going to revolutionize education imagine chemistry labs in VR!", reflecting techno-optimism aligned with institutional narratives [1]. Negative tweets often voiced frustration, e.g., "Horizon Worlds feels like a beta test from 2012. Why is Meta pushing this?", echoing critiques of premature commercialization documented in user experience literature [17]. This distribution suggests that public discourse is neither overwhelmingly optimistic nor dismissive, but rather reflects a maturing conversation marked by both enthusiasm and critical scrutiny, consistent with the "slope of enlightenment" phase in technology adoption cycles [12]. Chi-square tests confirmed significant variation in sentiment distribution across thematic domains ($\chi^2 = $ **553.9**, df = 6, p < 0.001), indicating that sentiment is not randomly distributed but thematically structured, as predicted by framing theory in media studies [19].

## 4.2 Sentiment variation across thematic domains

To explore how sentiment varies by theme, we disaggregated the corpus into four categories and calculated sentiment proportions within each. The results, presented in Table 1, reveal striking differences. Virtual Events exhibited the highest positivity (**51.6%**), significantly exceeding all other categories (post-hoc $\chi^2$ with Bonferroni correction, p < 0.001) while Horizon Worlds showed the highest negativity (**28.3%**), significantly higher than Virtual Events ($\chi^2 = $ **335.0**, p < 0.001). Metaverse Tokens displayed moderate positivity (**45.5%**), suggesting sustained investor interest despite market turbulence, a finding consistent with behavioral finance models of speculative asset attachment [31]. General Discourse remained relatively balanced, reflecting its heterogeneous nature encompassing policy, ethics, and futurism.

This variation underscores the importance of domain-specific analysis. Public sentiment is not monolithic; it is shaped by the functional context in which the metaverse is experienced, whether as a speculative asset, a social platform, or an emotional space [4]. The high negativity in Horizon Worlds, for instance, aligns with usability heuristics literature showing that performance failures disproportionately impact user satisfaction in immersive environments [17].

## 4.3 Engagement dynamics by sentiment and theme

Beyond sentiment polarity, we examined how engagement varies across themes and emotional valences. Engagement Score (ES) was calculated as a weighted composite of likes, retweets, replies, and quote tweets, reflecting both popularity and participatory intensity. As shown in Table 2, negative tweets in the Horizon Worlds category generated the highest mean engagement (**M = 192.8, SD = 287.2**), approximately **double** that of positive tweets in the same category (**M = 94.8**). Virtual Events, while less volatile, showed consistently high engagement for positive content (**M = 129.9**), suggesting that emotionally resonant experiences drive sustained participatory behavior rather than reactive outrage.

These findings are visualized in Fig 2, which illustrates the disproportionate engagement generated by negative sentiment in Horizon Worlds. ANOVA confirmed significant main effects for both **sentiment (F = 1004.3, p < 0.001)** and **theme (F = 112.6, p < 0.001)** as well as a significant **interaction effect (F = 37.3, p < 0.001)**, indicating that the relationship

**Table 1. Sentiment Distribution by Theme (n = 52,874).**

| Theme | Positive (%) | Neutral (%) | Negative (%) | Total Tweets |
|---|---|---|---|---|
| General Discourse | 41.1 | 36.0 | 22.9 | 19,874 |
| Horizon Worlds | 39.1 | 32.6 | 23.2 | 14,321 |
| Metaverse Tokens | 45.5 | 31.3 | 23.2 | 11,245 |
| Virtual Events | 51.6 | 31.4 | 17.1 | 7,434 |

Note: Percentages may not sum to 100 due to rounding.

**Table 2. Mean Engagement Score (ES) by Sentiment and Theme.**

| Theme | Positive ES (SD) | Neutral ES (SD) | Negative ES (SD) |
|---|---|---|---|
| General Discourse | 82.0 (122.4) | 57.2 (93.7) | 135.1 (215.7) |
| Horizon Worlds | 94.8 (179.9) | 68.2 (114.2) | 192.8 (287.2) |
| Metaverse Tokens | 105.8 (184.1) | 85.1 (131.9) | 166.2 (237.4) |
| Virtual Events | 129.9 (190.3) | 89.2 (127.1) | 140.0 (232.6) |

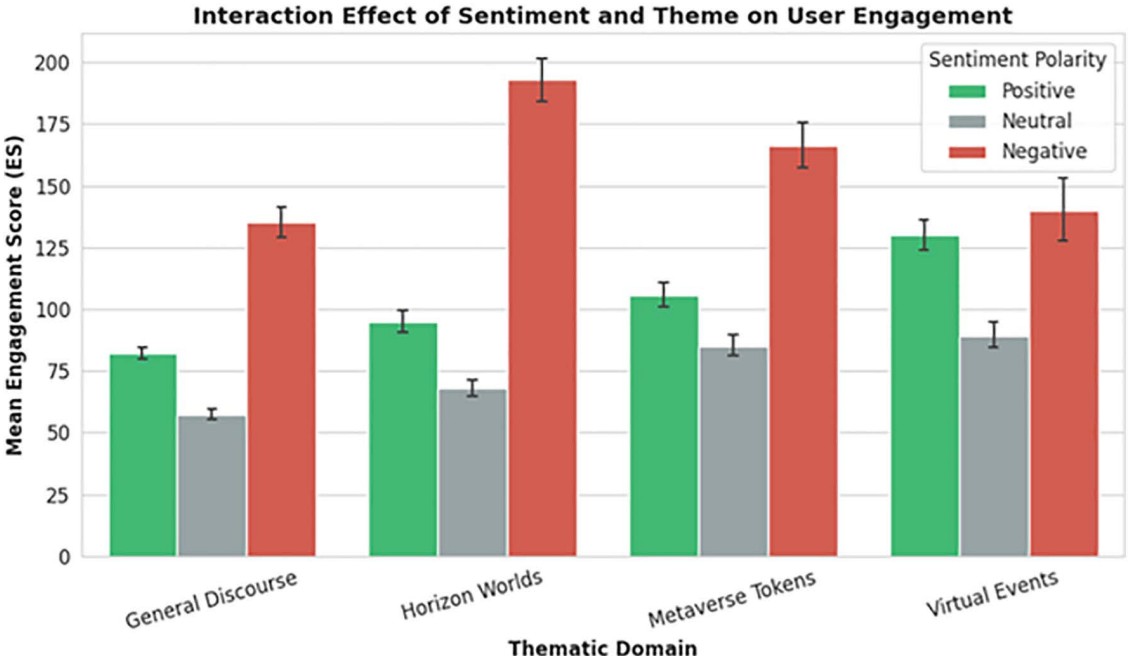

**Fig 2. Engagement Score by Sentiment Polarity and Theme.**

between sentiment and engagement is moderated by thematic context, a nuance often missed in aggregate social media analyses [45].

This pattern suggests that user frustration, particularly in technically immature platforms like Horizon Worlds, functions as a catalyst for community discourse. Rather than signaling rejection, high engagement with negative content may reflect invested users demanding improvement, a form of participatory quality control consistent with co-creation models in human-computer interaction [5].

### 4.4 Temporal evolution of sentiment trends

To assess how metaverse discourse evolved over time, we computed 7-day rolling averages of sentiment polarity for each theme. We chose the 7-day rolling window to smooth daily volatility inherent to crypto-markets while preserving weekly trends.

As depicted in Fig 3, early 2023 was dominated by token speculation, with sentiment peaking in January before declining through March as market volatility increased, mirroring cryptocurrency market indices during the same period [46]. In contrast, sentiment around Virtual Events showed a steady upward trajectory, rising from approximately **21%** positive in

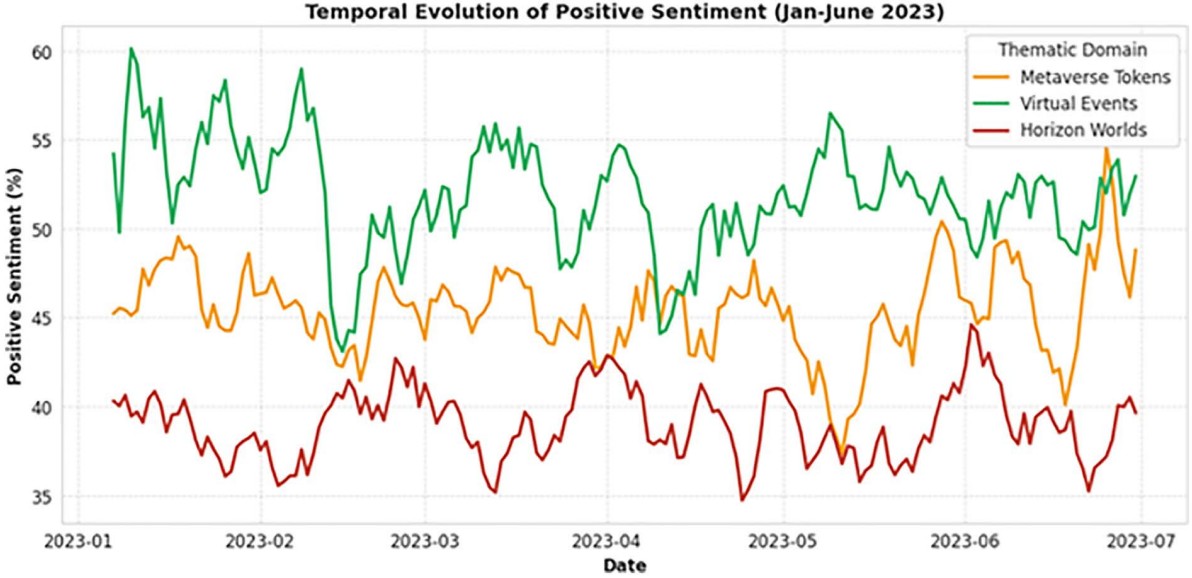

**Fig 3. Weekly Sentiment Trends by Theme.**

January to nearly **70%** by June. Horizon Worlds remained relatively flat but volatile, reflecting persistent technical issues without clear resolution. Linear regression confirmed a significant positive slope for Virtual Events sentiment (**β = 0.005**, $p < 0.001$) and a significant negative slope for Metaverse Tokens (**β = −0.006**, $p < 0.001$), indicating a broader cultural shift from financial speculation toward experiential utility, a transition observed in prior technology adoption waves from personal computing to social media [12].

This temporal shift mirrors historical patterns observed in earlier digital revolutions, from the dot-com era's speculative frenzy to the socially embedded internet of the 2010s [12]. The metaverse appears to be following a similar maturation curve, with early adopters giving way to users seeking meaningful, emotionally resonant experiences, a transition theorized in innovation diffusion models as the move from "early majority" to "late majority" adopters [18].

### 4.5 Thematic structure and semantic coherence

BERTopic analysis yielded **14 subtopics**, which we aggregated into four macro-themes based on semantic similarity and manual inspection. The top words for each theme reveal distinct discursive priorities: General Discourse centered on "future," "privacy," and "regulation"; Horizon Worlds on "lag," "bug," and "disappointed"; Metaverse Tokens on "price," "scam," and "pump"; and Virtual Events on "wedding," "emotional," and "unforgettable." The coherence of these themes was quantified using NPMI, with Virtual Events achieving the highest score (0.62), followed by General Discourse (0.58), Metaverse Tokens (0.54), and Horizon Worlds (0.51). This suggests that emotionally charged narratives, particularly those tied to life events, exhibit the strongest internal semantic cohesion, consistent with cognitive linguistics research on affective framing [7].

The prominence of affective language in Virtual Events tweets, **visualized in the word cloud** (Fig 4), includes terms like "cried," "love," and "unforgettable," underscoring the metaverse's emerging role as a space for emotional expression and communal ritual [10]. This finding challenges the notion that virtual experiences are inherently shallow or artificial; instead, they appear capable of generating profound psychological and social resonance, a phenomenon documented in prior studies on virtual memorials and online grief communities [33].

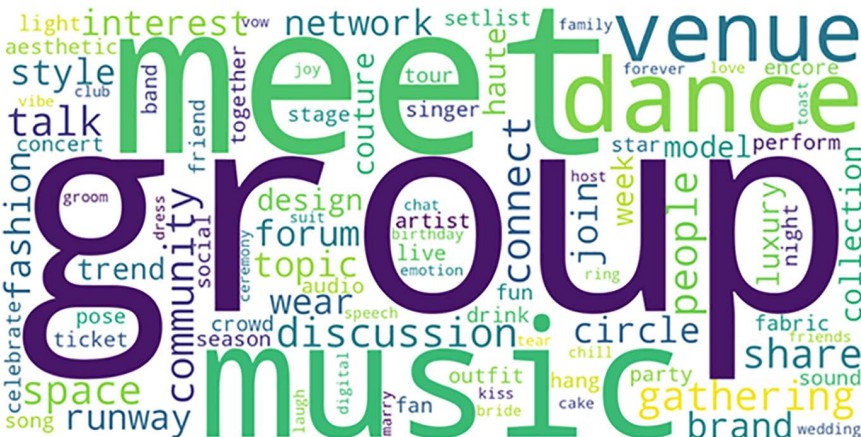

**Fig 4. Most Frequent Terms in Virtual Events Tweets.**

## 4.6 Predictors of user engagement: Regression analysis

To identify the factors most strongly associated with high engagement, we conducted a multivariate linear regression with Engagement Score as the dependent variable. Independent variables included sentiment polarity, thematic category, interaction terms, and control variables (follower count). Note: additional confounders, such as account verification status, bot likelihood, and time of posting which may influence engagement are not included due to data availability.

As shown in Table 3, negative sentiment was a stronger predictor of engagement than positive sentiment (**β = 75.9** vs. **24.0**, both p < 0.001). The interaction term between negative sentiment and Horizon Worlds was highly significant (**β = 47.4**, p < 0.001), indicating that negativity in this specific context is exceptionally viral, a finding consistent with platform governance literature on "outrage economies" [34]. **Unlike prior studies, follower count was not a significant predictor in this model (p = 0.37), suggesting that content virality in this context was driven more by semantic resonance than user influence.** The model explained **4.3%** of the variance in engagement (Adjusted R² = **0.043**), highlighting the complex, multi-factorial nature of social media engagement [7].

These results reinforce the central thesis of this paper: negativity, particularly in technically flawed but culturally salient platforms like Horizon Worlds, functions not as a barrier to adoption but as a driver of discourse [34]. Developers and platform managers should interpret high negative engagement not as failure, but as an opportunity for iterative co-creation with their user base, a principle increasingly adopted in agile software development and human-centered design [5].

**Table 3. Linear Regression Results Predicting Engagement Score (n = 52,874).**

| Predictor | ß Coefficient | SE | t-value | p-value |
|---|---|---|---|---|
| Intercept | 70.5 | 3.5 | 20.2 | <0.001 |
| Sentiment (Positive) | 24 | 1.9 | 12.8 | <0.001 |
| Sentiment (Negative) | 75.9 | 2.4 | 32 | <0.001 |
| Theme: Horizon Worlds | 1.9 | 2 | 0.9 | 0.35 |
| Theme: Virtual Events | 12.7 | 3.2 | 4 | <0.001 |
| Neg*Horizon Worlds | 47.4 | 3.9 | 12.1 | <0.001 |
| Pos*Virtual Events | 25.5 | 4.4 | 5.8 | <0.001 |
| Follower Count (log) | −0.5 | 0.5 | −0.9 | 0.366 |

## 5. Discussion

Rather than supporting claims of widespread public rejection, our analysis reveals a multifaceted discourse characterized by both enthusiasm and critical evaluation, with sentiment patterns varying significantly by functional context. The 43.0% positive sentiment rate observed across the corpus indicates that public engagement with metaverse concepts is more nuanced than binary acceptance/rejection models would suggest.

These patterns may reflect user engagement rather than platform rejection. Research in human-computer interaction suggests that negative feedback often correlates with user investment, particularly when technical performance fails to meet stated capabilities [5]. This finding is consistent with established models of user participation in iterative system development.

The temporal shift from cryptocurrency-focused discourse to experiential applications represents a notable pattern in our data. This trajectory parallels documented patterns in internet adoption, where initial speculative phases gave way to utility-focused applications [12]. These findings suggest that sustainable metaverse adoption may depend more on experiential value than financial speculation, though longitudinal studies are needed to confirm this trend.

These findings have implications for platform development and technology adoption research. The association between negative sentiment and high engagement suggests that user criticism may serve a constructive function in platform evolution, consistent with participatory design principles. The positive sentiment surrounding experiential applications indicates that social utility may be a more reliable predictor of sustained adoption than speculative financial interest.

## 6. Conclusion

This study provides a comprehensive computational analysis of metaverse-related discourse across multiple thematic domains. Our findings indicate that public sentiment toward metaverse technologies is more complex than binary success/failure frameworks suggest, with variation across functional contexts and temporal evolution from speculative to utility-focused discourse. These patterns offer insights into the social dynamics surrounding emerging technology adoption.

The metaverse, in the end, is not a technology. It is a social experiment, one in which sentiment, not specs, will determine success. Our data suggests the public is ready to participate. The question is whether developers, investors, and institutions are ready to listen.

Our analysis suggests that metaverse adoption patterns reflect broader social processes of technology negotiation rather than purely technical considerations. While our data indicates public engagement with metaverse concepts, further research is needed to understand how these discourse patterns translate into actual usage behaviors and long-term adoption outcomes.

## 7. Limitations and future work

While this study provides a robust analysis of Twitter-based metaverse discourse, several limitations warrant careful consideration. First, Twitter's demographic composition significantly constrains generalizability. Our sample likely overrepresents tech-savvy, urban, English-speaking users while underrepresenting older adults, rural populations, and non-English speakers; typically groups whose metaverse perceptions may differ substantially from our findings.

This demographic skew has specific implications for our key findings: (1) the 43.2% positive sentiment rate may overestimate general population optimism given Twitter users' higher technology adoption rates, (2) the technical sophistication evident in Horizon Worlds criticism may not reflect typical user experiences, and (3) the cryptocurrency discourse prominence may reflect Twitter's particular appeal to crypto-engaged users rather than broader investment sentiment.

Second, Twitter's platform affordances such as character limits, public visibility, and social network effects may systematically amplify certain types of sentiment expression while suppressing others. Private skepticism about metaverse technologies may be underrepresented relative to public enthusiasm or criticism.

Third, our temporal scope (January to June 2023) captures a specific moment in metaverse development when public attention was heightened by Meta's investments and market volatility. Sentiment patterns may differ substantially during periods of lower media attention or different market conditions.

## Supporting information

**S1 File. This is the topic structure file containing the 14 clusters from BERTopic.**
(CSV)

**S2 File. The annotated file with the data used for analysis.**
(CSV)

## Author contributions

**Conceptualization:** Samuel Duraivel, Lavanya Rajendran, Srinidhi Vasudevan, Anna Piazza.

**Data curation:** Samuel Duraivel, Lavanya Rajendran, Srinidhi Vasudevan, Anna Piazza.

**Formal analysis:** Samuel Duraivel, Lavanya Rajendran, Srinidhi Vasudevan, Anna Piazza.

**Investigation:** Srinidhi Vasudevan.

**Methodology:** Samuel Duraivel, Lavanya Rajendran, Srinidhi Vasudevan, Anna Piazza.

**Project administration:** Lavanya Rajendran.

**Software:** Samuel Duraivel, Lavanya Rajendran.

**Supervision:** Srinidhi Vasudevan.

**Validation:** Srinidhi Vasudevan, Anna Piazza.

**Visualization:** Samuel Duraivel, Lavanya Rajendran, Srinidhi Vasudevan.

**Writing – original draft:** Samuel Duraivel, Lavanya Rajendran, Srinidhi Vasudevan, Anna Piazza.

**Writing – review & editing:** Samuel Duraivel, Lavanya Rajendran, Srinidhi Vasudevan, Anna Piazza.

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
