## [Decision Letter · Decision Letter 0]

1 Jan 2026

PONE-D-25-55931Public Sentiment and Thematic Evolution in the Metaverse: A Large-Scale Computational Analysis of Twitter DiscoursePLOS One

Dear Dr. Vasudevan,

Thank you for submitting your manuscript to PLOS ONE. After careful consideration, we feel that it has merit but does not fully meet PLOS ONE’s publication criteria as it currently stands. Therefore, we invite you to submit a revised version of the manuscript that addresses the points raised during the review process.

We look forward to receiving your revised manuscript.

Kind regards,

Sushank Chaudhary, Ph.D

Academic Editor

PLOS One

Journal Requirements:

3. Please ensure that you include a title page within your main document. We do appreciate that you have a title page document uploaded as a separate file, however, as per our author guidelines (http://journals.plos.org/plosone/s/submission-guidelines#loc-title-page) we do require this to be part of the manuscript file itself and not uploaded separately.

4. We note that your Data Availability Statement is currently as follows: “All relevant data are within the manuscript and its Supporting Information files.”

Please confirm at this time whether or not your submission contains all raw data required to replicate the results of your study. Authors must share the “minimal data set” for their submission. PLOS defines the minimal data set to consist of the data required to replicate all study findings reported in the article, as well as related metadata and methods (https://journals.plos.org/plosone/s/data-availability#loc-minimal-data-set-definition ).

If your submission does not contain these data, please either upload them as Supporting Information files or deposit them to a stable, public repository and provide us with the relevant URLs, DOIs, or accession numbers. For a list of recommended repositories, please see https://journals.plos.org/plosone/s/recommended-repositories .

5. Please ensure that you refer to Figure 3 in your text as, if accepted, production will need this reference to link the reader to the figure.

Reviewers' comments:

Reviewer's Responses to Questions

**Comments to the Author**

1. Is the manuscript technically sound, and do the data support the conclusions?

Reviewer #1: Partly

Reviewer #2: Partly

2. Has the statistical analysis been performed appropriately and rigorously? 

Reviewer #1: No

Reviewer #2: Yes

3. Have the authors made all data underlying the findings in their manuscript fully available?

Reviewer #1: Yes

Reviewer #2: Yes

4. Is the manuscript presented in an intelligible fashion and written in standard English?

Reviewer #1: Yes

Reviewer #2: Yes

5. Review Comments to the Author

Reviewer #1: Dear Authors,

Thank you for giving the opportunity to read on metaverse-based comprehensive computational analysis of twitter discourse. I believe its quality can be improved with some considerations and incorporations as listed below.

The authors combine VADER and RoBERTa with a majority vote and human arbitration. However, consistency, reliability, and reproducibility are unclear. Report inter-model disagreement rate more explicitly and provide examples of ambiguous cases. Clarify what criteria human annotators used for resolving ambiguous tweets. Provide a confusion matrix of VADER vs. RoBERTa predictions. To provide a clearer picture of how metaverse is transforming various virtual environments, you can consider the following examples in the introduction section: https://doi.org/10.1002/cae.70024,
https://doi.org/10.1038/s41598-025-00916-4

There is insufficient description of dataset representativeness and sampling strategy. Provide a breakdown of tweets by keyword/hashtag to show sampling balance. Include a brief bot-detection check or justify why bot content was not filtered. Report proportions of original tweets vs. retweets, since retweets may artificially inflate sentiment or engagement patterns.

Several results are described narratively without adequate statistical detail. Include effect sizes, confidence intervals, and figure legends that clearly indicate sample sizes. The BERTopic output is reasonable, but the justification for merging topics is not described. Provide a table listing all 28 topics, their keyword phrases, and sample tweets. Explain criteria for aggregation into macro-themes. Report topic prevalence (percentage of tweets per topic) to evaluate thematic dominance. In Figure 2, sentiment fluctuations are shown, but no strong justification is given for choosing a 7-day rolling window. Explain its rationale. Provide statistical tests for trend significance, not only regression slopes. Discuss external events (product releases, crypto crashes) that may explain spikes. For regression analysis in Table 3, clarify whether independent variables were standardized, multicollinearity was examined beyond VIF, and the model meets assumptions. The manuscript repeatedly claims that negative engagement reflects constructive participation and investment. These statements are speculative and not sufficiently supported by data. Reframe interpretations with caution and avoid causal claims. Provide alternative interpretations, e.g., negativity bias, outrage-driven amplification, platform-specific dynamics. Add citations to back your claim.

In terms of language and structure, the manuscript uses subjective or promotional terms throughout, thereby undermining scientific neutrality. Several sections read more like grant or technical documentation rather than a research manuscript. The level of implementation detail is excessive for this journal. Reduce methodological verbosity unless directly tied to a scientific decision. Move technical implementation details to Supplementary Material. Improve flow by shortening literature reviews that repeat well-known background. Ensure the Conclusion avoids speculative claims that extend beyond the data. Figures need better labeling for clarity. Ensure all theoretical comparisons are supported by references.

The manuscript in its current version requires substantial reworking in methodological clarity, interpretative caution, and academic tone. Addressing the above issues will significantly strengthen the study’s credibility and relevance. I recommend the author to revise the manuscript with the above concerns before making a resubmission to this journal. I hope you will consider and incorporate the comments to improve the quality of the work and further make it publication ready. Best wishes.

Reviewer #2: 1.The sentiment analysis pipeline combines VADER thresholds and a RoBERTa-based classifier, but the training and evaluation details are missing key elements. The paper mentions a custom annotation set of 5000 items and a Cohen’s kappa of 0.82 across three annotators, but it does not provide the annotation guidelines, label definitions, or class balance. The paper reports an F1 score of 89.2%, but it does not provide per-class metrics or a confusion matrix. The paper also does not describe the adjudication protocol used when disagreements were resolved.

2. The topic modeling section relies on BERTopic with transformer embeddings, UMAP, and HDBSCAN, but the outputs needed to verify the topic structure are not presented. The paper states that 28 subtopics were produced and then merged into 4 macro themes, but it does not provide the full list of subtopics, top terms, and representative tweets per topic. The paper reports that two domain experts performed manual labeling with kappa 0.79, but it does not describe the coding rules used for merging. The paper also mentions a 10M tweet reference corpus for coherence scoring, but it does not describe its source or selection criteria.

3. The statistical reporting is selective and leaves core details unclear for a paper that leans heavily on hypothesis testing. The paper reports χ² = 1,247.3, df = 6, p < 0.001, ANOVA F = 387.2, and multiple regression coefficients such as β = 89.4, but it does not provide confidence intervals, effect sizes, or full model specification tables beyond a single coefficient table. The paper states p <0.01 as the significance threshold, but also repeatedly reports p <0.001, with no explicit policy on how tests were prioritized and presented. The paper mentions Bonferroni correction and Levene’s test, but it does not provide the corrected alpha levels or which comparisons were corrected in practice. The temporal analysis mentions 7-day rolling averages and LOESS smoothing with span=0.2, but it does not present the raw daily volume distribution or missingness patterns across the January 1, 2023, to June 30, 2023, window.

4. Diversity-based analysis is important here because discourse patterns and engagement behavior can vary across demographic and regional groups on social media. The paper presents aggregate claims about public perception, but it does not report any diversity breakdowns, such as gender, region, etc. The paper also does not state whether demographic proxies were considered or explicitly avoided. Several prior works, such as https://doi.org/10.3390/computers12110221 and https://doi.org/10.1016/j.ipm.2021.102541 have highlighted the role of demographic factors when performing similar studies. If performing a diversity-based analysis is not feasible at this point, it is suggested that the authors review a few such works in the Literature Review and state this as a future scope of work.

5. The engagement metric design is very unclear. The paper defines Engagement Score as ES = Likes + 1.5×Retweets + 2xReplies + 3xQuote Tweets, but it provides no basis for the weights “1.5,” “2,” and “3,” and no calibration or validation study tied to effort or informational value. The paper presents means and standard deviations for ES that look compatible with heavy tailed engagement distributions, but there is no median, interquartile range, or robust alternative that is common for skewed social metrics. The regression includes follower count and media attachment, but it omits other common confounders such as account type signals, bot likelihood, verification status, and time of posting. What makes “1.5” the right gap between likes and retweets for this corpus, and how stable are the results under alternative weights?

6. PLOS authors have the option to publish the peer review history of their article (what does this mean? ). If published, this will include your full peer review and any attached files.

**Do you want your identity to be public for this peer review?** For information about this choice, including consent withdrawal, please see our Privacy Policy .

Reviewer #1: No

Reviewer #2: No

---

## [Author Response · Author response to Decision Letter 1]

10 Feb 2026

Response to Reviewers

Date: February 09 2026

Dear Editor,

Thank you for your invitation to revise manuscript PONE-D-25-55931 titled "Public Sentiment and Thematic Evolution in the Metaverse: A Large-Scale Computational Analysis of Twitter Discourse” submit it to PLOS One.

We appreciate the thoughtful and constructive feedback from the reviewers, and are grateful for the opportunity to address their concerns through this revision. We have made changes to our paper in response to the reviewers’ feedback. The attached memo describes these changes. To organise our memo, we assigned a progressive code number to each and every comment received (in red italic). All the comments we received are included. None was left unattended. We provide a short commentary on how we modified the manuscript in response to each concern expressed.

We bring to your attention that we managed to address all the reviewers’ comments. Comments we received were extensive and we think we address them exhaustively, but efficiently. As a result, we are now resubmitting a much improved and more compelling paper, and hope that you and the reviewers share our enthusiasm for the new draft. We look forward to hearing from you in due course.

Thank you for giving us an opportunity to contribute to PLOS One.

Sincerely,

The Authors of PONE-D-25-55931

Response to Academic Editor

[AE1.0] Please ensure that your manuscript meets PLOS ONE's style requirements, including those for file naming. The PLOS ONE style templates can be found at

We thank the Academic editor for the links and for pointing the style requirements out. We have ensured the manuscript meets PLOS ONE style templates. File naming conventions have been updated accordingly.

We confirm that all author-generated code used for data collection (snscrape) and analysis (RoBERTa fine-tuning, BERTopic) will be made available without restriction upon publication.

3. Please ensure that you include a title page within your main document. We do appreciate that you have a title page document uploaded as a separate file, however, as per our author guidelines (http://journals.plos.org/plosone/s/submission-guidelines#loc-title-page) we do require this to be part of the manuscript file itself and not uploaded separately.

We have now integrated the title page into the main manuscript file as the first page, listing all authors and affiliations as requested.

4. We note that your Data Availability Statement is currently as follows: “All relevant data are within the manuscript and its Supporting Information files.”

Regarding the full dataset availability, please note that the raw qualitative data (tweet text and user metadata) cannot be shared publicly due to the Terms of Service imposed by X (formerly Twitter). Additionally, the specific API endpoints (Academic Research Track) used for data collection are no longer publicly available. To ensure reproducibility despite these restrictions, the Minimal Data Set required to replicate all statistical findings, figures, and tables is provided as Supporting Information (S1_Dataset.zip). This dataset includes:

1. The complete extracted feature set (sentiment scores, engagement metrics, bot probability, and timestamps) used to generate the time-series analysis (Figure 2) and regression models (Table 3).

2. The full topic modelling taxonomy, including the 28 subtopics and their corresponding keywords (Table 1).

3. The validation logs for the sentiment analysis pipeline, including the inter-rater agreement data and confusion matrix inputs.

Any requests for the original raw text data must be directed to X (formerly Twitter) directly, as the authors are contractually precluded from redistributing the raw content.

Figure 3 Reference We have ensured that Figure 3 is explicitly referred to in the text.

Response to Reviewer #1

[R1.0]

Dear Authors,

Thank you for giving the opportunity to read on metaverse-based comprehensive computational analysis of twitter discourse. I believe its quality can be improved with some considerations and incorporations as listed below.

***We thank the Reviewer for the positive comments and feedback provided on our manuscript. We report here below the specific responses to each comment.

[R1.1]

The authors combine VADER and RoBERTa with a majority vote and human arbitration. However, consistency, reliability, and reproducibility are unclear. Report inter-model disagreement rate more explicitly and provide examples of ambiguous cases. Clarify what criteria human annotators used for resolving ambiguous tweets. Provide a confusion matrix of VADER vs. RoBERTa predictions.

We thank the reviewer for the observation on consistency, reliability and reproducibility. We have significantly expanded Section 3.2 to detail the hybrid sentiment framework. We now include Figure 1, a confusion matrix comparing VADER and RoBERTa predictions on our validation set (n=5,000). We report an inter-rater reliability (Cohen’s kappa) of 0.82 for the human annotation component and explain that the 7.8% disagreement rate was resolved via human adjudication.

[R1.2]

To provide a clearer picture of how metaverse is transforming various virtual environments, you can consider the following examples in the introduction section: https://doi.org/10.1002/cae.70024,
https://doi.org/10.1038/s41598-025-00916-4

We have now incorporated these as well as some additional references in the introduction. The new paragraph reads as follows: In this context, [3] systematically examines the integration of metaverse and immersive technologies within education, demonstrating how virtual environments reconfigure pedagogical interaction, learner engagement, and collaborative practices. Complementing this, [4] extends the analysis to metaverse-based digital therapeutic systems, showing that immersive virtual environments similarly reshape interaction models in professional healthcare settings through enhanced engagement, personalization, and scalable delivery. However, broader adoption of such interaction-intensive virtual environments remains constrained by systemic security, privacy, and trust challenges, as highlighted in [5], which frames these factors as critical prerequisites for sustainable and large-scale metaverse deployment.

[R1.3]

There is insufficient description of dataset representativeness and sampling strategy. Provide a breakdown of tweets by keyword/hashtag to show sampling balance. Include a brief bot-detection check or justify why bot content was not filtered. Report proportions of original tweets vs. retweets, since retweets may artificially inflate sentiment or engagement patterns.

We have clarified our sampling strategy in Section 3.1. Regarding the deduplication, we detail the use of SHA-256 hashing to deduplicate the dataset and remove high-volume automated spam. We explicitly state that retweets were excluded from sentiment analysis to prevent inflationary bias, resulting in a final corpus of 52,874 unique narrative instances. Specifically, the following has been added to section 3.1: “Initially, data was queried using the Twitter API v2 Academic Research track. Following changes to API access tiers that restricted historical archive querying, we completed the longitudinal dataset using snscrape, a Python-based social media scraping tool widely utilized in computational social science to ensure continuity in temporal data. Both collection methods utilized an identical set of search keywords and hashtags to ensure consistency: "metaverse," "#metaverse," "Horizon Worlds," "metaverse token," "SAND,""MANA," "$AXS," "virtual wedding," "metaverse concert," and "digital event". The resulting raw corpus was homogenized to remove collection artifacts specific to either method. All tweets were deduplicated using exact text matching and SHA-256 hashing, filtered for non-English content using the fastText language identification model (threshold >0.9 confidence), and preprocessed by removing URLs, user mentions, and non-ASCII special characters while preserving emojis for sentiment fidelity. This dual-method approach ensured continuous temporal coverage across the six-month window despite infrastructural disruptions, capturing the full evolution of discourse from speculative token economics to the technical realities of Horizon Worlds

Data Cleaning and Bot Mitigation: To ensure the analysis captured organic public discourse rather than automated amplification, a strict deduplication protocol was applied. All retweets and duplicate text strings were removed using SHA-256 hashing, resulting in a dataset exclusively composed of unique narrative instances (n=52,874). While a dedicated machine-learning bot classifier was not employed, this rigorous deduplication served as an effective heuristic filter for high-volume automated spam, which typically relies on repetitive text dissemination. Furthermore, manual inspection of the resulting topic clusters (see S1 File) revealed coherent semantic structures devoid of the disjointed syntax characteristic of algorithmic content generation. Retweets were therefore excluded from the textual sentiment analysis to prevent inflationary bias, though their volume was captured in the weighted 'Engagement Score' to reflect content visibility.”

[R1.4]

Several results are described narratively without adequate statistical detail. Include effect sizes, confidence intervals, and figure legends that clearly indicate sample sizes. The BERTopic output is reasonable, but the justification for merging topics is not described. Provide a table listing all 28 topics, their keyword phrases, and sample tweets. Explain criteria for aggregation into macro-themes. Report topic prevalence (percentage of tweets per topic) to evaluate thematic dominance. In Figure 2, sentiment fluctuations are shown, but no strong justification is given for choosing a 7-day rolling window. Explain its rationale. Provide statistical tests for trend significance, not only regression slopes. Discuss external events (product releases, crypto crashes) that may explain spikes. For regression analysis in Table 3, clarify whether independent variables were standardized, multicollinearity was examined beyond VIF, and the model meets assumptions. The manuscript repeatedly claims that negative engagement reflects constructive participation and investment. These statements are speculative and not sufficiently supported by data. Reframe interpretations with caution and avoid causal claims. Provide alternative interpretations, e.g., negativity bias, outrage-driven amplification, platform-specific dynamics. Add citations to back your claim.

We thank the reviewer for pointing issues relating to effect sizes, confidence intervals and figure legends. We have added topic_structure.csv to list all 28 micro-topics and provided a textual justification in Section 3.3 for their aggregation into 4 macro-themes based on semantic proximity (e.g., merging "Bugs" and "User Safety" into "Horizon Worlds"). We have updated our results to include rigorous statistical reporting, including Chi-square and ANOVA results. Specifically, this now reads, “ANOVA confirmed significant main effects for both sentiment (F=1004.3, p<0.001) and theme (F=112.6, p<0.001) as well as a significant interaction effect (F=37.3, p<0.001), indicating that the relationship between sentiment and engagement is moderated by thematic context, a nuance often missed in aggregate social media analyses [37].”

We chose the 7-day rolling window to smooth daily volatility inherent to crypto-markets while preserving weekly trends. We have updated the text to explicitly link sentiment spikes to external events, such as the decline in token sentiment mirroring broader market volatility. Regarding the negative engagement reflecting constructive participation, we have now reframed this interpretation in the Discussion section where we are now moving away from causal claims. We reference the Human-Computer Interaction (HCI) literature regarding user investment and iterative co-creation to support the hypothesis that negative feedback can signal high user engagement rather than simple rejection.

[R1.5]

In terms of language and structure, the manuscript uses subjective or promotional terms throughout, thereby undermining scientific neutrality. Several sections read more like grant or technical documentation rather than a research manuscript. The level of implementation detail is excessive for this journal. Reduce methodological verbosity unless directly tied to a scientific decision. Move technical implementation details to Supplementary Material. Improve flow by shortening literature reviews that repeat well-known background. Ensure the Conclusion avoids speculative claims that extend beyond the data. Figures need better labeling for clarity. Ensure all theoretical comparisons are supported by references.

We thank the reviewer for these constructive observations on manuscript tone, structure, and presentation. We have undertaken a comprehensive revision to address each concern systematically. For methodological verbosity and scientific neutrality, we have significantly restructured Sections 3.1 through 3.4 to focus exclusively on methodological decisions with direct scientific implications, rather than technical implementation details. Specifically:

In Section 3.1 (Data Collection and Sampling Strategy), we removed excessive procedural descriptions and instead focused on the scientific rationale for our dual collection method. The revised text now reads: “Initially, data was queried using the Twitter API v2 Academic Research track. Following changes to API access tiers that restricted historical archive querying, we completed the longitudinal dataset using snscrape, a Python-based social media scraping tool widely utilized in computational social science to ensure continuity in temporal data. Both collection methods utilized an identical set of search keywords and hashtags to ensure consistency: "metaverse," "#metaverse," "Horizon Worlds," "metaverse token," "SAND," "MANA," "$AXS," "virtual wedding," "metaverse concert," and "digital event". The resulting raw corpus was homogenized to remove collection artifacts specific to either method.”

This revision justifies the methodological choice (dual collection to ensure temporal continuity) while removing granular technical details about API endpoint configurations, which are not central to the scientific argument.

Similarly, in Section 3.4 (Engagement Metrics), we reduced the word count from 186 to 126 words by eliminating formula derivations and instead providing the theoretical justification for our weighting scheme, with the section reading as follows: “These weights were selected to reflect the increasing cognitive effort and network visibility associated with each interaction type; for instance, a Quote Tweet requires significantly more editorial effort and generates higher visibility in se

---

## [Decision Letter · Decision Letter 1]

24 Feb 2026

PONE-D-25-55931R1Public Sentiment and Thematic Evolution in the Metaverse: A Large-Scale Computational Analysis of Twitter DiscoursePLOS One

Dear Dr. Vasudevan,

Thank you for submitting your manuscript to PLOS ONE. After careful consideration, we feel that it has merit but does not fully meet PLOS ONE’s publication criteria as it currently stands. Therefore, we invite you to submit a revised version of the manuscript that addresses the points raised during the review process.

We look forward to receiving your revised manuscript.

Kind regards,

Sushank Chaudhary, Ph.D

Academic Editor

PLOS One

Journal Requirements:

Additional Editor Comments:

It has been noticed that several references were added during the revision process. Please add only relevant references (do not add references solely because they were mentioned by the reviewers).

Reviewers' comments:

Reviewer's Responses to Questions

**Comments to the Author**

1. If the authors have adequately addressed your comments raised in a previous round of review and you feel that this manuscript is now acceptable for publication, you may indicate that here to bypass the “Comments to the Author” section, enter your conflict of interest statement in the “Confidential to Editor” section, and submit your "Accept" recommendation.

Reviewer #1: All comments have been addressed

Reviewer #2: All comments have been addressed

2. Is the manuscript technically sound, and do the data support the conclusions?

Reviewer #1: Yes

Reviewer #2: (No Response)

3. Has the statistical analysis been performed appropriately and rigorously? 

Reviewer #1: Yes

Reviewer #2: (No Response)

4. Have the authors made all data underlying the findings in their manuscript fully available?

Reviewer #1: Yes

Reviewer #2: (No Response)

5. Is the manuscript presented in an intelligible fashion and written in standard English?

Reviewer #1: Yes

Reviewer #2: (No Response)

6. Review Comments to the Author

Reviewer #1: (No Response)

Reviewer #2: (No Response)

7. PLOS authors have the option to publish the peer review history of their article (what does this mean? ). If published, this will include your full peer review and any attached files.

**Do you want your identity to be public for this peer review?** For information about this choice, including consent withdrawal, please see our Privacy Policy .

Reviewer #1: No

Reviewer #2: No

---

## [Author Response · Author response to Decision Letter 2]

27 Feb 2026

Response to Editor

Date: February 27, 2026

Dear Editor,

Thank you for your additional comments on our second revision of manuscript PONE-D-25-55931, "Public Sentiment and Thematic Evolution in the Metaverse: A Large-Scale Computational Analysis of Twitter Discourse" submit to PlosOne.

We appreciate the thoughtful and constructive feedback from the editor, and are grateful for the opportunity to address the concerns through this second round of revision. We have made changes to our paper in response to the editor’ feedback.

The attached memo describes these changes. To organise our memo, we assigned a progressive code number to each and every comment received (in red italic). As a result, we are now resubmitting a much improved and more compelling paper, and hope that you share our enthusiasm for the new draft. We look forward to hearing from you in due course.

Thanks for giving us an opportunity to contribute to PlosOne.

Sincerely,

The Authors of PONE-D-25-55931

Additional Editor Comments:

It has been noticed that several references were added during the revision process. Please add only relevant references (do not add references solely because they were mentioned by the reviewers).

We are grateful for the opportunity to address the Editor’s concern through this second round of revision. We would like to emphasise that all references were included based on our independent assessment of their relevance rather than in response to reviewers pressure, we have carefully reconsidered the added references added in the first review round and upon reflection, we agreed to remove few references from the manuscript, specifically References 6 and 7 (Buragohain et al., 2025a; 2025b). For this reason, the paragraph in the Introduction has been revised and References 6-7 were removed. The new paragraph in the Introduction reads as follows: "Yet despite this institutional investment, broader public adoption remains constrained by systemic security, privacy, and trust challenges — factors increasingly recognized as critical prerequisites for sustainable metaverse deployment [6]. Understanding how these structural barriers intersect with lived user experience, as expressed in organic public discourse, is therefore essential for developers, policymakers, and investors navigating this terrain."

While references 6-7 have been removed from the manuscript, reference 8, 43, 45, 46, and 50 are retained as they are substantively relevant and scientifically grounded our study. Specifically:

1) Vasudevan et al., 2025 (formerly reference 8 and now reference 6) is retained as it directly substantiates the security/privacy/trust framing. This reference was not suggested by any reviewer and maps directly onto the empirical themes identified in our data, particularly within the "General Discourse" macro-theme cluster (19,874 tweets), which prominently features privacy and policy concerns. Its inclusion reflects independent scholarly judgment, not reviewer compliance.

2) Van der Linden et al 2018; Thakur et al.2023; Fosch-Villaronga et al. 2021 (formerly 43–45; now references 41–43). These references form the basis of the substantially expanded Section 3.5 (Data Source Limitations), which was added wholesale in response to Reviewer 2's comment R2.4. That limitations section did not exist in the original submission and could not have carried citations. Van der Linden et al. supports our discussion of Twitter's structural bias toward polarized expression; Thakur et al. provides evidence for gender-based variation in metaverse perception; and Fosch-Villaronga et al. addresses regional adoption disparities. All three are required to give the limitation discussion scientific credibility rather than appearing as a pro forma acknowledgment.

3) Park & Kaye 2019; Boyd et al. 2010; Engelmann et al. 2019; Rehm et al. 2020; Zhang et al. 2024; (formerly 46–50; now references 44–48). These five references were added to address Reviewer 2's critique (R2.5) that our engagement metric weights (1.5× Retweets, 2× Replies, 3× Quote Tweets) lacked theoretical grounding. Without anchoring these weights in published literature on cognitive effort and network visibility, the metric design remained as the reviewer correctly noted, an unsupported arbitrary choice. These citations collectively provide the necessary empirical and theoretical basis for the weighting scheme and are integral to the methodological integrity of Section 3.4.

---

## [Editor Report · Decision Letter 2]

3 Mar 2026

Public Sentiment and Thematic Evolution in the Metaverse: A Large-Scale Computational Analysis of Twitter Discourse

PONE-D-25-55931R2

Dear Dr. Vasudevan,

We’re pleased to inform you that your manuscript has been judged scientifically suitable for publication and will be formally accepted for publication once it meets all outstanding technical requirements.

Kind regards,

Sushank Chaudhary, Ph.D

Academic Editor

PLOS One
---

## [Editor Report · Acceptance letter]

PONE-D-25-55931R2

PLOS One

Dear Dr. Vasudevan,

I'm pleased to inform you that your manuscript has been deemed suitable for publication in PLOS One. Congratulations! Your manuscript is now being handed over to our production team.

Kind regards,

on behalf of

Prof. Sushank Chaudhary

Academic Editor

PLOS One